# Design of Photocatalytic Functional Coatings Based on the Immobilization of Metal Oxide Particles by the Combination of Electrospinning and Layer-by-Layer Deposition Techniques

**Xabier Sandua** [1,2,*], **Pedro J. Rivero** [1,2], **Joseba Esparza** [3], **José Fernández-Palacio** [3], **Ana Conde** [4] and **Rafael J. Rodríguez** [1,2]

1 Engineering Department, Campus Arrosadía s/n, Public University of Navarre, 31006 Pamplona, Spain; pedrojose.rivero@unavarra.es (P.J.R.); rafael.rodriguez@unavarra.es (R.J.R.)
2 Institute for Advanced Materials and Mathematics (INAMAT2), Campus Arrosadía s/n, Public University of Navarre, 31006 Pamplona, Spain
3 Centre of Advanced Surface Engineering, AIN, 31191 Cordovilla, Spain; jegorraiz@ain.es (J.E.); jfpalacio@ain.es (J.F.-P.)
4 National Center for Metallurgical Research (CENIM-CSIC), Gregorio del Amo Avenue 8, 28040 Madrid, Spain; a.conde@cenim.csic.es
* Correspondence: xabier.sandua@unavarra.es

**Abstract:** This work reports the design and characterization of functional photocatalytic coatings based on the combination of two different deposition techniques. In a first step, a poly(acrylic acid) + β-Cyclodextrin (denoted as PAA+ β-CD) electrospun fiber mat was deposited by using the electrospinning technique followed by a thermal treatment in order to provide an enhancement in the resultant adhesion and mechanical resistance. In a second step, a layer-by-layer (LbL) assembly process was performed in order to immobilize the metal oxide particles onto the previously electrospun fiber mat. In this context, titanium dioxide (TiO$_2$) was used as the main photocatalytic element, acting as the cationic element in the multilayer LbL structure. In addition, two different metal oxides, such as tungsten oxide (WO$_3$) and iron oxide (Fe$_2$O$_3$), were added into PAA anionic polyelectrolyte solution with the objective of optimizing the photocatalytic efficiency of the coating. All of the coatings were characterized by scanning electron microscopy (SEM) and atomic force microscopy (AFM) images, showing an increase in the original fiber diameter and a decrease in roughness of the mats because of the LbL second step. The variation in the wettability properties from a superhydrophilic surface to a less wettable surface as a function of the incorporation of the metal oxides was also observed by means of water contact angle (WCA) measurements. With the aim of analyzing the photocatalytic efficiency of the samples, degradation of methyl blue (MB) azo-dye was studied, showing an almost complete discoloration of the dye in the irradiated area. This study reports a novel combination method of two deposition techniques in order to obtain a functional, homogeneous and efficient photocatalytic coating.

**Keywords:** photocatalysis; metal oxides; electrospinning; layer-by-layer; methyl blue

## 1. Introduction

Similarly to photosynthesis, photocatalysis is a type of photoreaction, where a catalyst adsorbs light, inducing chemical reactions that can have a huge impact in environmental applications [1]. Among them, air [2–4] and water purification [5–7] or pollutant and bacterial removal [8–12] are the most promising applications. The most widely used photocatalytic material since the 1970s has been titanium dioxide (TiO$_2$), due to its excellent performance and easy availability [13]. This compound is activated under UV irradiation, carrying out photocatalytic chemical reactions [14].

This chemical phenomenon occurs due to the transfer of electrons from the valence band to the conduction band [15]. TiO$_2$ can be crystallized in three different structures,

named rutile, anatase and brookite. Anatase and rutile both have photocatalytic activity. However, due to the high rate of electron-hole recombination, rutile is less effective than anatase [16]. Anatase has a band gap of 3.2 eV, and it is usually activated under UV light with a wavelength under 388 nm, providing enough energy for the electron transfer through the band gap [17]. The photoinduced electron-holes react with substances such as $O_2$, $H_2O$ and $OH^-$ groups, producing reactive oxygen species (ROS) that are responsible for the degradation of organic pollutants [18]. In addition, thanks to its excellent photocatalytic behavior, $TiO_2$ presents an optimal interaction with water media, due to its hydrophilic character [19]. This photoinduced hydrophilicity provides self-cleaning and antifogging features to the surface where the $TiO_2$ is present. Both photocatalytic and hydrophilic effects involve different photochemical reactions, thus both effects can occur at the same time and in the same surface area [20]. These unique properties and numerous practical applications associated with crystalline titanium dioxide are stimulating research on improving the existing materials and developing new alternative titanium dioxide synthetic chemical routes [21,22]. In this context, multiple strategies are being developed that will greatly enhance the structural stability as well as photocatalytic activity of $TiO_2$ with potential industrial applications (e.g., hydrogen evolution reaction, overall water splitting, photodegradation of organic pollutants) [23,24].

It is important to note that not only $TiO_2$ has photocatalytic behavior; many other metal oxides also have the property of inducing chemical reactions by interacting with light, such as ZnO, which has been widely used [25]. The combination of $TiO_2$ with other metal oxides has been also studied by evaluating the dynamics of photogenerated charge carriers by using specific semiconductors that exhibit well-defined heterojunctions and new properties [26,27]. In this context, the combination of $TiO_2$ with other metal oxides such as $WO_3$ [28] and $Fe_2O_3$ [29] has been also reported in order to enhance the photocatalytic behavior of the coating, narrowing the resultant band gap of $TiO_2$ [30]. As a result, an increase in visible light absorption by $TiO_2$ can be obtained, and the resultant heterojunction can effectively improve the lifetime of the photogenerated electron-hole pairs [31]. However, not all metal oxides interact in the same way with light. $TiO_2$ has a peak of light absorbance in the UV range (380 nm), whereas the peaks for $WO_3$ and $Fe_2O_3$ are in the range of visible spectra (460 nm and 620 nm, respectively) [13]. This fact is important in order to determine the light power supply parameters for the photocatalytic performance [32]. Moreover, $TiO_2$ also can be combined with metallic nanoparticles (mostly silver or gold), making possible visible light-induced photocatalysis [33,34].

In this work, a combination of two deposition techniques was applied in order to achieve functional photocatalytic surfaces. As the base element, an electrospun fiber mat of poly(acrylic acid) (PAA) was deposited into the substrate thanks to the electrospinning process. This versatile deposition technique is capable of obtaining ultrathin fibers from a viscous solution, forming a porous coating on the sample and providing functional surfaces for a wide range of applications [35]. Operational parameters of the solution (polymer concentration, molecular weight, viscosity, surface tension, conductivity) as well as processing parameters (applied voltage, flow rate, tip-collector distance) have to be optimized in order to obtain electrospun fibers with a desired morphology [36–38]. Once these parameters are set, a high-voltage supply is applied to the tip of the syringe where the solution is located and a cone shape filament structure is projected, yielding the electrospun coating [39,40].

The sample covered with the PAA-based electrospun fiber mat is then submitted into a layer-by-layer (LbL) immersive assembly in order to immobilize photocatalytic metal oxide particles into the coating. This deposition process is based on alternate immersions of the sample in different charged solutions [41,42]. Accurate control of the operational parameters (material concentration in solution, immersion time, washing conditions, number of bilayers and solution pH) is achieved in order to control the film's growth [43]. In order to obtain a homogenous distribution of the metal oxide particles, their dispersion within the solution has to be considered [44–47]. The combination of both deposition techniques

provides new features to the functional surface, compared with the coatings that could be achieved separately with these techniques. The matrix obtained by the electrospinning process makes possible a homogeneous structure with good adhesion onto the reference substrate, whereas the LbL assembly provides good affinity towards the electrospun fibers with good control over the thickness of the coatings and dispersion of the metal oxide particles [48–50].

Finally, components of wastewater plants can suffer corrosion degradation over time. This issue can lead to a failure of the process if this problem is not taken into account. For this reason, this paper also presents a preliminary evaluation of the role played by the coating in the corrosion performance of stainless steel substrate. Moreover, there is also an interesting new research field that studies the possible photocathodic protection, where photocatalytic materials (such as $TiO_2$), can produce electrons under light irradiation that are transferred to the protected metal in order to reduce metal self-corrosion potential [51]. The interest in photocathodic protection is continuously growing due to its low cost, non-toxic and environmentally friendly application [52,53]. Although this paper is not directly concerned with this effect, it analyzes the electrochemical protection provided by these combined electrospun fibers with multilayer LbL structure coatings to a standard material employed in pipelines, pumps and tanks (AISI 304 stainless steel) [54] as a preliminary step for further research on their photocathodic protection ability.

## 2. Experimental Section

### 2.1. Materials

For the electrospinning process solution, poly(acrylic acid) (PAA; Mw ≈ 450,000) and β-cyclodextrin (β-CD, purity 98%) were diluted in ethanol (99%). For the layer-by-layer process, titanium oxide ($TiO_2$, pure anatase nanopowder < 25 nm) was diluted in ultrapure water, acting as polycation (positive charged solution), whereas an aqueous solution of poly(acrylic acid) (PAA; Mw ≈ 250,000) acted as polyanion (negative charged solution), and at the same time, this polyelectrolyte also acted as an encapsulating agent of tungsten oxide ($WO_3$, <25 μm) and iron oxide ($Fe_2O_3$, <5 μm) particles. The azo-dye, methylene blue (MB), was used. All chemicals were obtained from Sigma-Aldrich (St. Louis, MO, USA). All reagents were used without any further purification, and ultrapure water with a resistivity of 18.2 MΩ·cm was employed as deionized water for the solutions.

The obtained functional coatings were deposited onto standard glass slides (75 × 50 mm) for complete characterization of their contact angles and morphological and photocatalytic properties. Finally, these coatings were also deposited onto metallic substrates (austenitic stainless steel AISI 304, 80 × 60 mm²) for characterization of the resultant corrosion behavior.

### 2.2. Deposition Techniques

Firstly, the reference substrate was deposited by electrospinning technique. In order to achieve the desired electrospun fiber coating, three main parameters were controlled: flow rate of the solution, applied voltage between the syringe tip and collector, and the tip–collector distance. By controlling them, the denominated "Taylor cone" was achieved, forming the coating fibers [55]. The solution employed in this deposition process contained 11.6 mL of ethanol as the solvent element, to which 0.8 g of PAA and 0.128 g of β-CD were added [56]. This solution was stirred for 24 h at room temperature until the solute was completely dissolved, leading to a homogeneous mixture. In order to fabricate the electrospun coating, this solution was placed in a syringe with 20-gauge needle with an inner diameter of 0.6 mm. The voltage applied between the syringe tip and the samples placed on an aluminum board connected to ground was 15 kV, whereas the distance between the tip and the samples was 15 cm. Finally, a flow rate of 1.3 mL/h was applied in the process. The same setup has been previously described elsewhere [56]. Once the electrospun fibers were deposited onto the reference substrates, all samples were treated at 180 °C for 40 min in order to promote chemical crosslinking and induce better stability of

the electrospun fiber mats. A schematic representation of the electrospinning process is shown in Figure 1a.

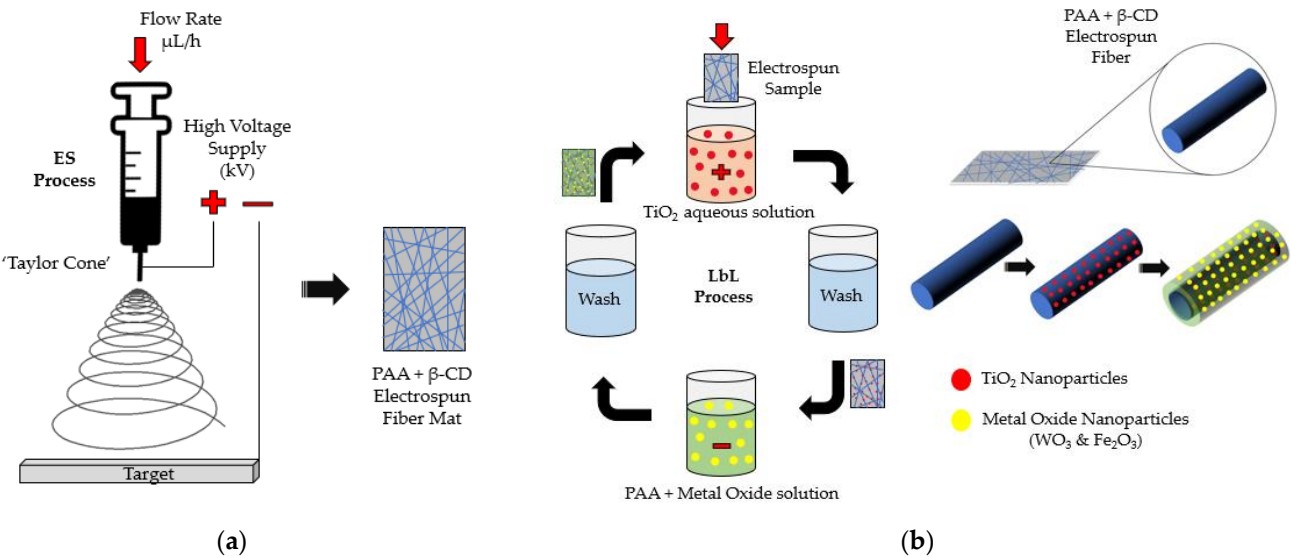

**Figure 1.** Schematic description of the fabrication of the thin films. (**a**) Electrospinning deposition technique; (**b**) Layer-by-layer assembly deposition technique.

For fabrication of the LbL multilayer structure onto the electrospun fiber mat, a weak polyelectrolyte, PAA, was employed as polyanion due to its property of negative charge [57]. As the polycation, an aqueous dispersion of $TiO_2$ was prepared ($TiO_2^+$), in which the metal oxide particles held a positive charge once the $TiO_2$ was introduced, stirred and sonicated in ultrapure water [58]. In the first monolayer, these photocatalytic particles adhered to the electrospun fiber mat due to the negative charge of the matrix. Then, a sequence of five $TiO_2^+$/PAA bilayers was carried out [50]. The $TiO_2^+$ polycation solution was prepared by diluting 250 mg $TiO_2$ nanoparticles in 250 mL of ultrapure water (10 mM) under stirring conditions for 2 h at room temperature, followed by an ultrasonication process for 30 min. Aqueous PAA was also diluted with a concentration of 671 mg in 250 mL of ultrapure water under vigorous stirring for 2 h at room temperature. Both solutions and intermediate ultrapure water solutions were adjusted to pH 2.0 [59].

Photocatalytic enhanced metal oxide particles with a molar concentration of 10 mM (724 mg $WO_3$ and 498 mg $Fe_2O_3$ in 250 mL of ultrapure water, respectively) were diluted into 250 mL of 10 mM PAA anionic solutions [60–62]. The anionic solutions with metal oxide particles were stirred for 2 h at room temperature and then submitted to ultrasonication for 30 min. Thus, PAA in combination with negative charged metal oxide particles were adhered to positive $TiO_2^+$ particles to produce the functional coating, as represented in Figure 1b.

### 2.3. Morphology, Wettability and Roughness Characterization

Water contact angles were measured using a Theta (Attension) optical tensiometer (CAM 100 KSV Instruments, Burlington, VT, USA). Sessile drops were recorded in fast mode with a trigger 5 s after the drop touched the surface. Contact angles were measured using the tangent algorithm drop profile fitting method. An average of 30 frames/s were used to calculate the contact angle for each drop. Each contact angle value is cited as an average of five measurements performed at different locations on the specimen surface.

Surface images were obtained by a field emission-scanning electron microscopy (FE-SEM Hitachi S3800, Tokyo, Japan), so that the fiber diameter could be visualized. This parameter was numerically measured by a confocal microscope (model S-mart, SENSOFAR

METROLOGY, Barcelona, Spain) with an objective of EPI 50X v35 for a final area of $340.03 \times 283.73 \ \mu m^2$.

Atomic force microscopy (AFM, Veeco Innova AFM, Veeco Instruments, Plainview, NY, USA) was also employed in order to study the roughness of the coatings. The AFM images were analyzed with Gwyddion software.

### 2.4. Photocatalytic Activity

Photodegradation of methylene blue (MB) in different samples was measured with the optical transmission setup shown in Figure 2, in order to study the photocatalytic efficiency of the functional coatings. The setup employed two types of power light sources. One provided light in the visible range, so that the methylene blue spectra could be displayed, and the photocatalytic activities of $WO_3$ and $Fe_2O_3$ particles could be activated in this range of the light spectrum. The other source provided light in the UV range (maximum spectra at $\lambda = 365$ nm), which was responsible for activating the photocatalytic activity of the $TiO_2$ particles [63]. Both light sources were connected to a bifurcated fiber, which guided both visible and UV light to the holder where the methylene blue sample was placed. Then, an optical fiber was attached into a UV-VIS spectrometer (Ocean Optics USB2000, Ocean Insight, FL, USA) that was also connected to a computer with the corresponding software, yielding reliable spectrum data between 200 and 800 nm. The photocatalytic study was conducted after immersion of the samples into 2.5 mM of methyl blue (MB) solution (azo-dye) for 2 h, showing a characteristic blue coloration (denoted as Methylene Blue sample in Figure 2) with the presence of two absorption bands at 625 nm and 675 nm (corroborated by UV-Vis spectra) [8]. Once the azo-dye was light-irradiated, the related peaks of the MB sample gradually decreased. All photocatalytic experiments were performed for a total time of 10 h, respectively.

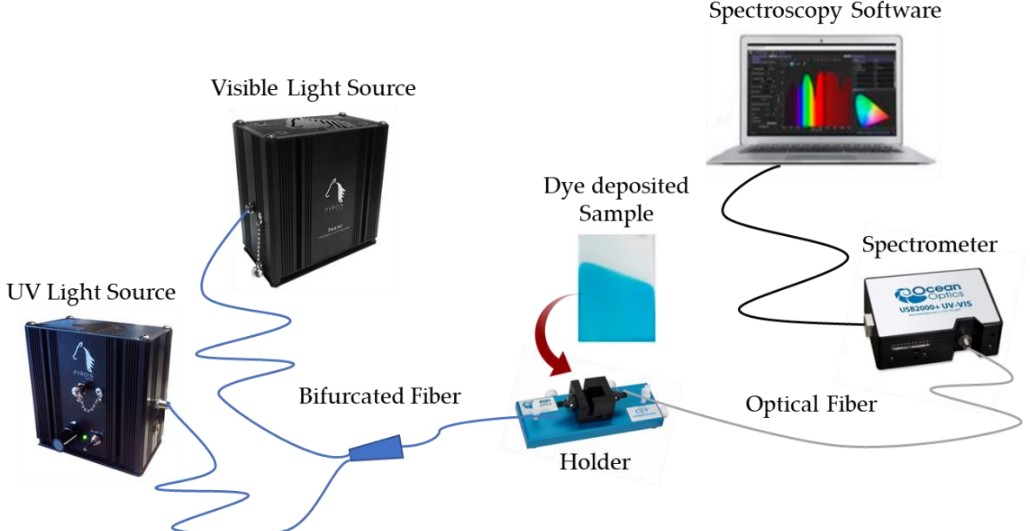

**Figure 2.** Scheme of the described setup for photocatalytic efficiency determination of the samples.

### 2.5. Electrochemical Measurements

Electrochemical characterization was performed by potentiodynamic curves (PC). The tests were performed in a three-electrode ($n = 3$) cell connected to a Gamry Reference 600 Potentiostat. Saline solution containing 35 g/L of NaCl was used as the electrolyte. The coated samples were used as the working electrode, with a 3 M KCl silver–silver chloride electrode (Ag/AgCl 3 M KCl) as reference electrode and a rolled platinum wire as counter electrode.

Before obtaining the PC curves, the open circuit potential (OCP) was recorded for 900 s. Subsequently, a cathodic potential step of $-0.3$ V (vs. OCP) was applied, and the anodic sweep was started at 0.16 mV/s until a current density limit of 0.25 mA/cm$^2$ was reached.

## 3. Results

This section is divided in three main subsections. Firstly, it describes the evolution of the water contact angle by adding the metal oxide particles. Secondly, an exhaustive analysis of the coating morphology is explained by using SEM and AFM images. Finally, the photocatalytic efficiency of the different samples as well as the electrochemical corrosion behavior are also studied. The reference sample (denoted as REF) and the different samples in this study (from S1 up to S3) are summarized in Table 1.

**Table 1.** Summary table of the analyzed samples.

| REF | S1 | S2 | S3 |
|:---:|:---:|:---:|:---:|
| PAA + β-CD | PAA + β-CD //TiO$_2$/PAA | PAA + β-CD //TiO$_2$/PAA + WO$_3$ | PAA + β-CD //TiO$_2$/PAA + Fe$_2$O$_3$ |

### 3.1. Contact Angle Measurement

It is well known how parameters such as interlayer penetration, roughness, chemical composition of polyelectrolytes and features of the outermost surface layer directly affect the wettability property of the functional coating [64]. Figure 3 shows the superhydrophilic effect associated with the functional hydroxyl groups of PAA in the initial electrospun fiber matrix (REF). In addition, it has to be considered that the water contact angle of a TiO$_2$ surface without external UV light source, according to literature, has a value of about 54° [65]. This wettability value was observed in the S1 coating (Figure 4), which had a contact angle of 52.8°. Hence, the immobilization of these particles made possible a change in wettability properties from superhydrophilic PAA to a hydrophilic surface after addition of the metal oxides.

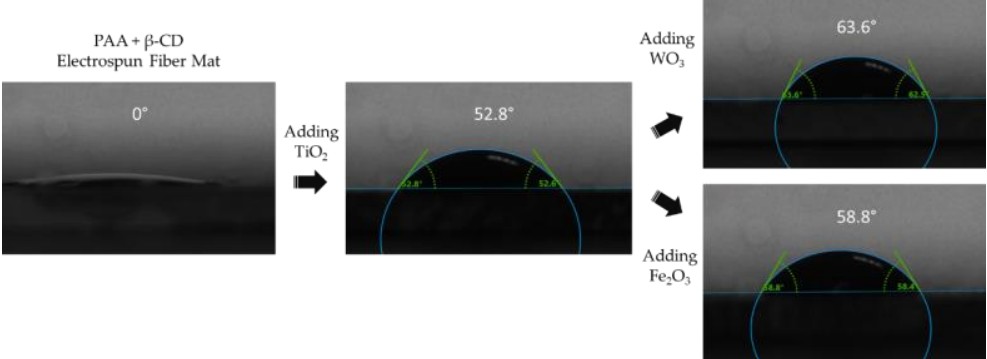

**Figure 3.** Water contact angle images for the analyzed samples.

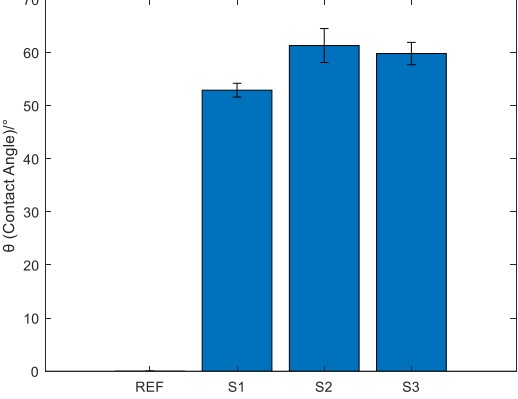

**Figure 4.** Water contact angle numeric measurements for the analyzed samples.

Like TiO$_2$, other photocatalytic metal oxides also present superhydrophilic performance under UV irradiation [66,67]. Nevertheless, their behavior without any external light source tends to be more hydrophobic. Thus, the hydrophobic performance of the surface is improved by adding metal oxides to the surface [64], such as Fe$_2$O$_3$ [68] and WO$_3$ [69]. Figure 4 shows the increase in contact angle after the addition of metal oxide particles. By adding WO$_3$, the contact angle increased by 10°, whereas the Fe$_2$O$_3$ addition increased the contact angle by 6° compared to the TiO$_2$/PAA coating. Thus, the change in contact angle was not significant between the samples containing metal oxides.

Figure 4 shows the numerical value graph for these measurements. It can be concluded that the addition of metal oxide particles to the coating increases the contact angle, obtaining less wettable surfaces from a super-hydrophilic one. This fact is due to the hydrophobic properties of the metal oxides.

### 3.2. Morphology and Roughness Characterization

Images of the samples obtained by SEM show the morphology of the electrospun fiber mats before and after the LbL process (see Figure 5). Figure 5a shows the uncoated PAA + β-CD electrospun fiber mat with the formation of uniform and homogenous electrospun fibers with a slight presence of beads [70]. Once layer-by-layer technique was applied with the corresponding metal oxide particles (Figure 5b–d), and the resultant fiber diameter of the samples was notably increased. In addition, the characteristic porosity of the electrospinning deposition [71] was decreased because of this increase in fiber diameter after the LbL process [72].

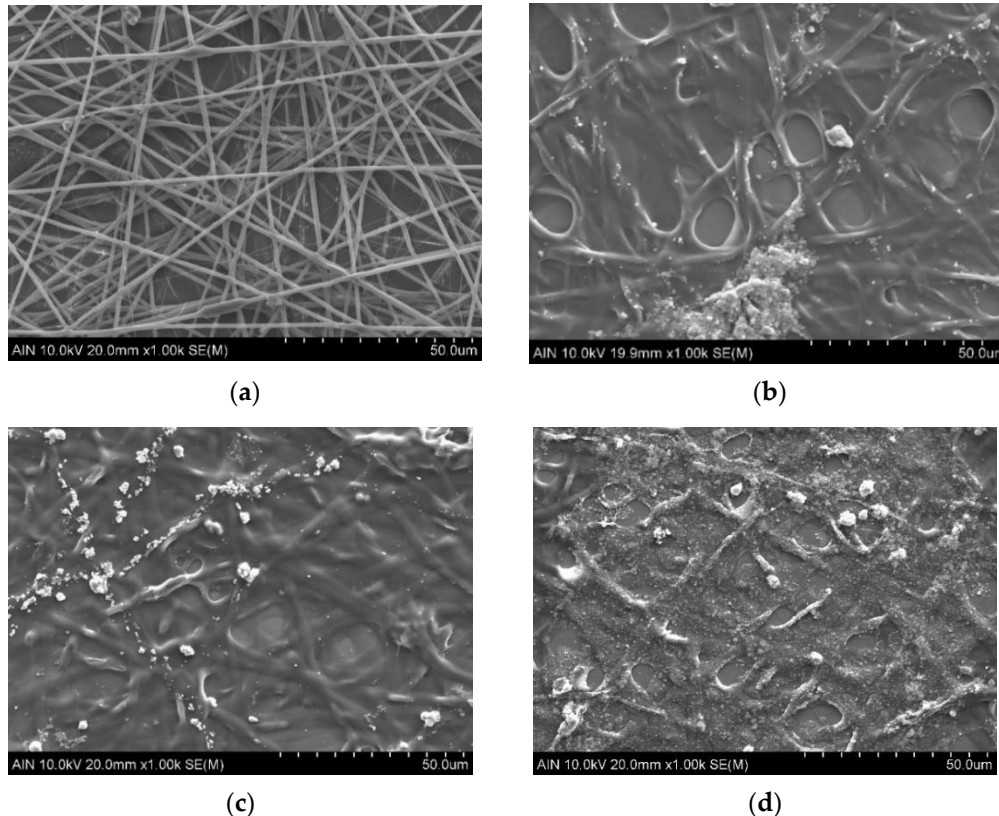

(**a**)　　　　　　　　　　　　　　　　　　　　　　　　　　(**b**)

(**c**)　　　　　　　　　　　　　　　　　　　　　　　　　　(**d**)

**Figure 5.** SEM images of the four analyzed samples: (**a**) PAA + β-CD; (**b**) PAA + β-CD//TiO$_2$/PAA; (**c**) PAA + β-CD//TiO$_2$/PAA + WO$_3$; (**d**) PAA + β-CD//TiO$_2$/PAA + Fe$_2$O$_3$.

Figure 6 presents SEM images as well as EDX analyses of samples 2, 3 and 4, respectively, in order to show the distribution of the metal oxide particles on the outer surface of the electrospun fibers. In all figures (Figure 6a–c), the green area corresponds to the titanium signal related to the TiO$_2$ nanoparticles, whereas the blue area corresponds to

tungsten related to $WO_3$ particles (Figure 6b) and iron (Figure 6c) related to $Fe_2O_3$ particles. After observing both SEM (Figure 5) and SEM/EDX (Figure 6) images, the presence of localized agglomeration sites of the metal oxide particles were observed on the electrospun fiber mats, and the resultant particle size distributions could be disturbed [73].

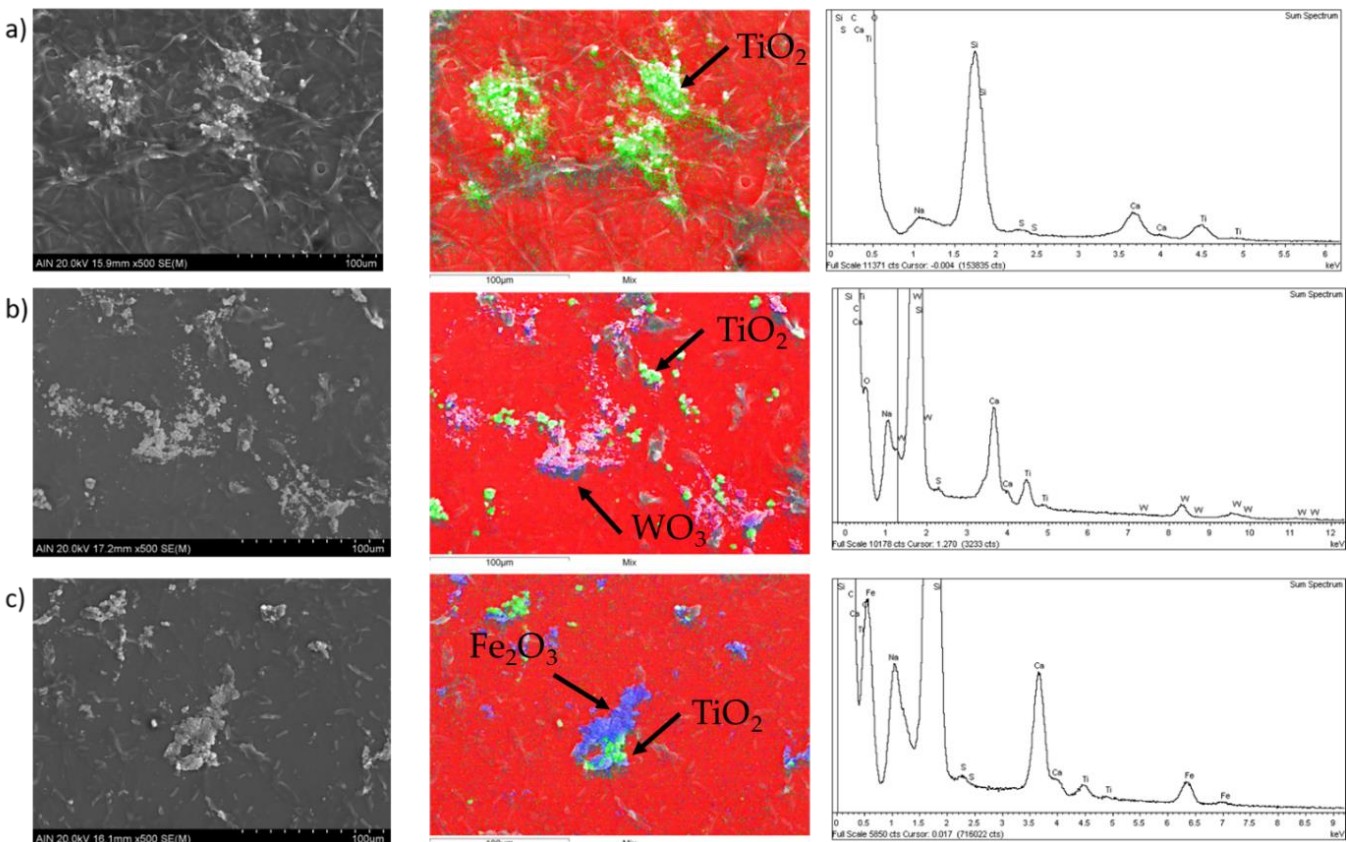

**Figure 6.** SEM/EDX images of the four analyzed samples: (**a**) PAA + β-CD//$TiO_2$/PAA; (**b**) PAA + β-CD//$TiO_2$/PAA + $WO_3$; (**c**) PAA + β-CD//$TiO_2$/PAA + $Fe_2O_3$.

Table 2 lists the changes in the fiber diameters of the different samples as the metal oxides were added into the coating. Hence, it can be concluded that samples S2 and S3, which contained the combination of two different metal oxide particles, had the greatest coarsening of fiber diameter.

**Table 2.** Variation in fiber diameters of the different samples analyzed.

|  | **REF** | **S1** | **S2** | **S3** |
|---|---|---|---|---|
| φ (μm) | 1.11 ± 0.12 | 1.49 ± 0.21 | 3.76 ± 0.42 | 3.23 ± 0.56 |

Finally, the addition of metal oxides into the coating also gave rise to agglomerations of the metal oxide particles into the functional surface [73]. Metal oxide particles tend to form agglomerates in aqueous media [74]. These particle agglomerates could be observed in the three coating systems that contained metal oxides (Figure 5b–d). Thus, an increase in the resultant roughness was produced in those areas, as shown by the AFM images (Figure 7), where the major roughness increase was due to the agglomeration of the metal oxide particles.

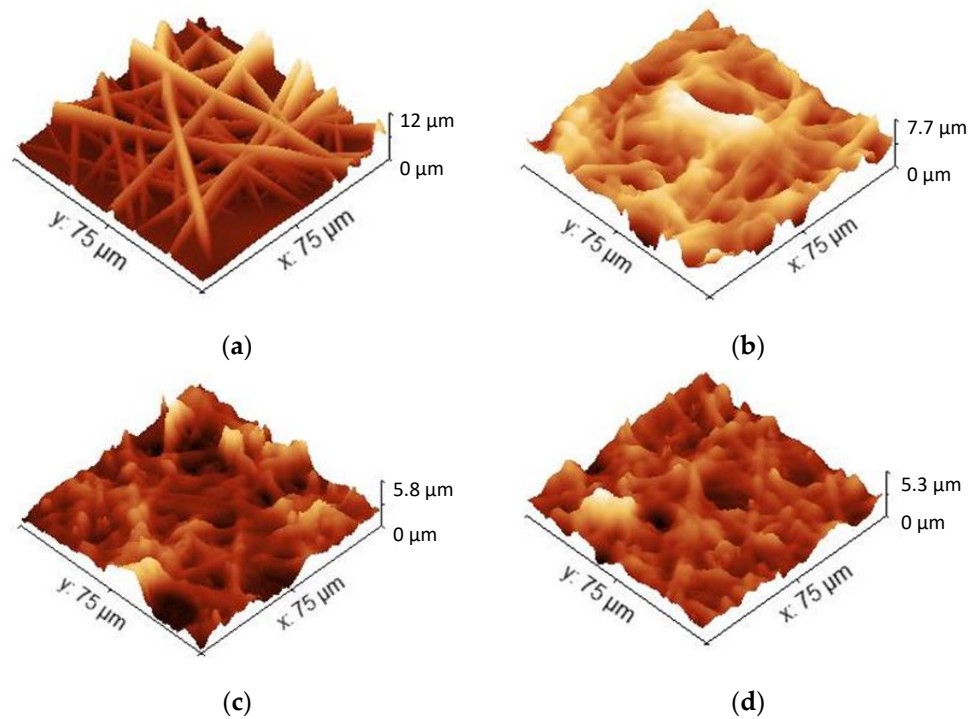

**Figure 7.** AFM 3D images of the four analyzed samples (**a**) PAA + β-CD; (**b**) PAA + β-CD//TiO$_2$/PAA; (**c**) PAA + β-CD//TiO$_2$/PAA + WO$_3$; (**d**) PAA + β-CD//TiO$_2$/PAA + Fe$_2$O$_3$.

Table 3 lists two different roughness parameters, average roughness (Ra) and root mean square roughness (Rq). The table data allow us to conclude that as the metal oxide particles were added, the roughness was decreased. The roughest sample corresponded to the electrospun fiber mat, without any metal oxides. As the TiO$_2$, WO$_3$ or Fe$_2$O$_3$ particles were added, the roughness decreased because of the increase in fiber diameter. Finally, Figure 8 shows that the thickness of samples increased as the metal oxides were added into the coating.

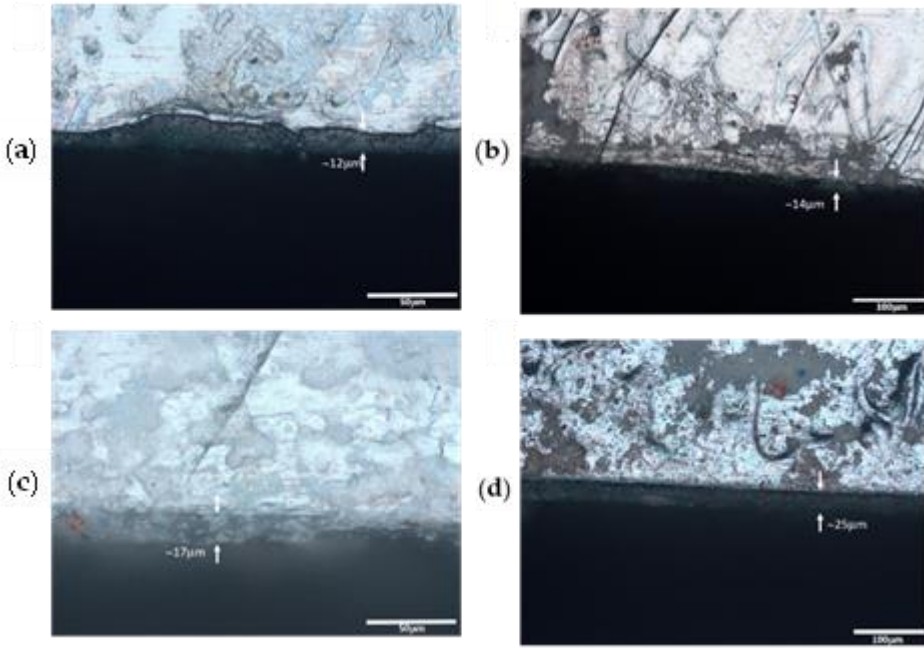

**Figure 8.** Average coating thicknesses for Ref (**a**), S1 (**b**), S2 (**c**) and S3 (**d**), respectively.

**Table 3.** Roughness parameters of different samples.

|  | REF | S1 | S2 | S3 |
|---|---|---|---|---|
| Ra (μm) | $0.58 \pm 0.16$ | $0.34 \pm 0.12$ | $0.12 \pm 0.02$ | $0.19 \pm 0.03$ |
| Rq (μm) | $0.76 \pm 0.13$ | $0.42 \pm 0.14$ | $0.15 \pm 0.01$ | $0.24 \pm 0.04$ |

### 3.3. Photocatalytic Efficiency

The photocatalytic behavior of the samples was tested by measuring the discoloration of MB. These degradation rates can be observed in Figure 9, showing the individual Gaussian components at two specific wavelength locations, 620 and 675 nm. In addition, the evolution of these individual bands over time is shown in Figure 10, showing a similar trend in these two wavelength positions. The photocatalytic performance of the electrospun fiber matrix of PAA + β-CD without metal oxide particles could be disregarded (Figure 9a) [75], allowing the conclusion that deposition without any photocatalytic particles did not contribute to the degradation of the colorant. Thus, the S1-coating system containing $TiO_2$ particles in contact with the MB dye (Figure 9b) showed a notable decrease in absorbance, leading to 65% degradation of the pollutant. As mentioned in the introduction, the combination of photocatalytic metal oxides enhanced the photodegradation of the functional coating [28–30]. For the S2-coating system combining $TiO_2$ and $WO_3$ oxides, the photodegradation of the azo-dye was improved by 10%, attaining a removal rate of 75% in a 10 h test (Figure 9c). This similarly occurred with the S3-coating system, leading to a methylene blue degradation of 85% in a 10 h test (Figure 9d). Hence, an improvement in the photodegradation rate of an azo-dye by means of the combination of $TiO_2$ with two different metal oxides was corroborated. In addition, observation of the fiber diameter measurements demonstrated that an increment in the resultant fiber diameter (mostly S2 and S3) led to a greater distribution of the photocatalytic particles, with a corresponding increase in the functional photocatalytic area of the samples [49,50].

It is important to note that the resultant $TiO_2$ photocatalysis can be improved by the addition of other photoreactive metal oxides that act as effective dopants for the $TiO_2$ photocatalysis [30]. Accordingly, the combination of $WO_3$ with $TiO_2$ particles produced a decrease in the band gap (Eg value), thus facilitating an enhancement in the photocatalytic activity under visible light. In addition, the combination of both $Fe_2O_3$ and $TiO_2$ reduced the resultant band gap of $TiO_2$, acting as a temporary trap of surface charges and increasing the photocatalytic activity under visible and solar light for dye degradation [76]. Comparison of these two metal oxide precursors showed that $Fe_2O_3$ had a narrower band gap (Eg = 2.2 eV) than WO3 (Eg = 2.6 eV), and due to this, the resultant photocatalytic response could be slightly better than that with $WO_3$ [77,78].

These results are in agreement with previous works, allowing the conclusion that mixing $TiO_2$ and $Fe_2O_3$ can achieve almost total discoloration in a few hours [56]. The durability of the photocatalytic activity of the samples was demonstrated in order to prove the reusability of the system [79]. After carrying out the first photocatalytic test, another two regeneration cycles were carried out. Samples were immersed in the same MB solution until the same peak intensities were achieved. Then, photocatalytic tests were conducted for 10 h in order to analyze the reusability of the resultant samples. All of the tests showed that the degradation of MB followed the same pattern and the photocatalytic activity was fully sustained. This is summarized in Figure 11, which shows the decrease in the maximum absorbance peak of MB in samples over time using a logarithmic scale [80].

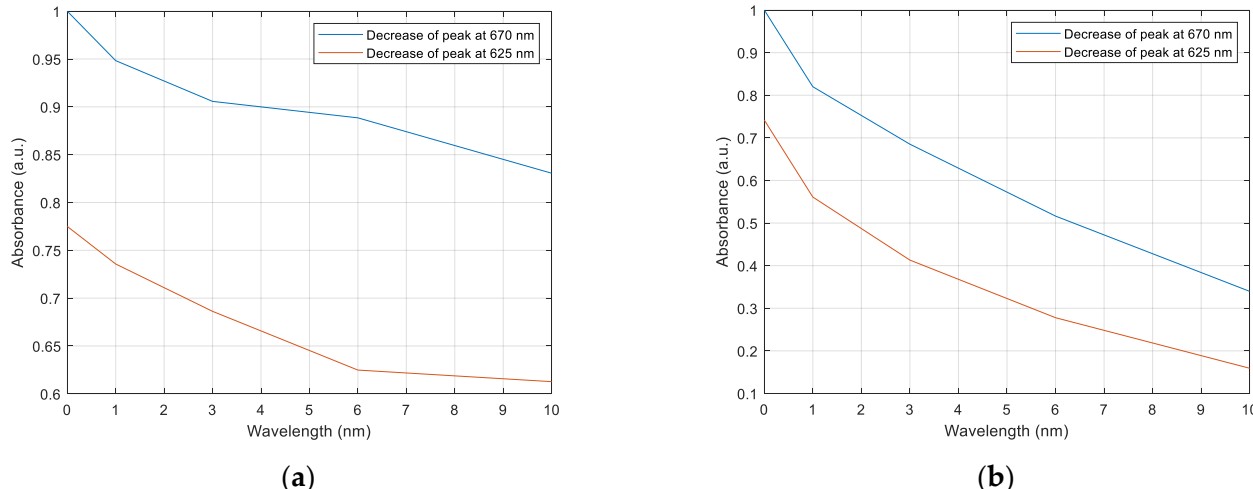

**Figure 9.** Photodegradation rates after total irradiation of the different samples for 10 h. (**a**) PAA + β-CD; (**b**) PAA + β-CD//TiO$_2$/PAA; (**c**) PAA + β-CD//TiO$_2$/PAA + WO$_3$; (**d**) PAA + β-CD//TiO$_2$/PAA + Fe$_2$O$_3$.

**Figure 10.** *Cont.*

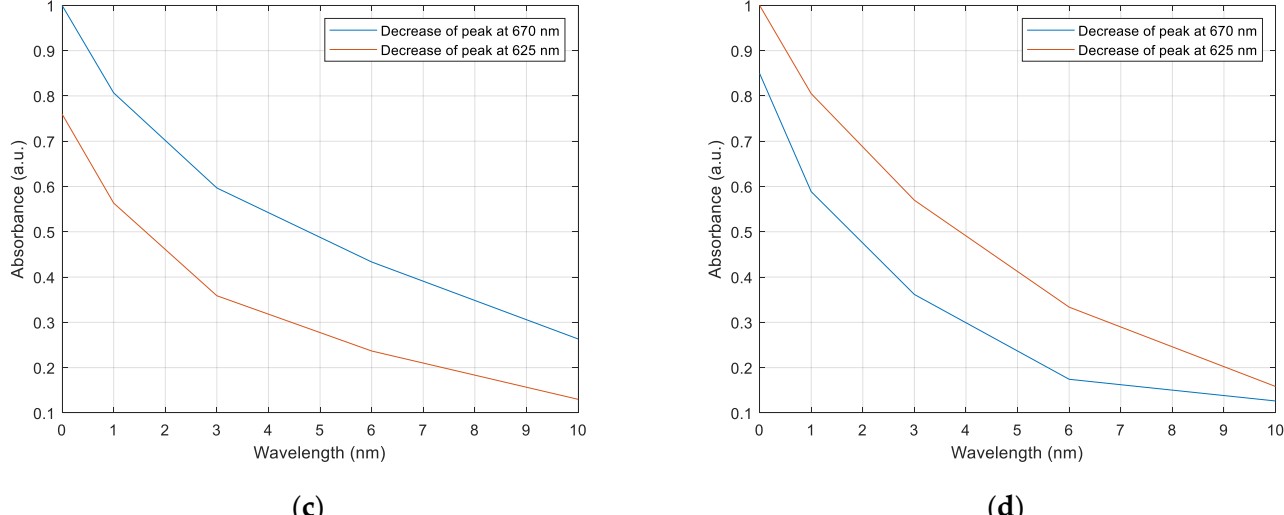

**Figure 10.** Photodegradation evolution rates of the individual Gaussian components over time at 620 and 675 nm of the different samples under total irradiation for 10 h. (**a**) PAA + β-CD; (**b**) PAA + β-CD//TiO$_2$/PAA; (**c**) PAA + β-CD//TiO$_2$/PAA + WO$_3$; (**d**) PAA + β-CD//TiO$_2$/PAA + Fe$_2$O$_3$.

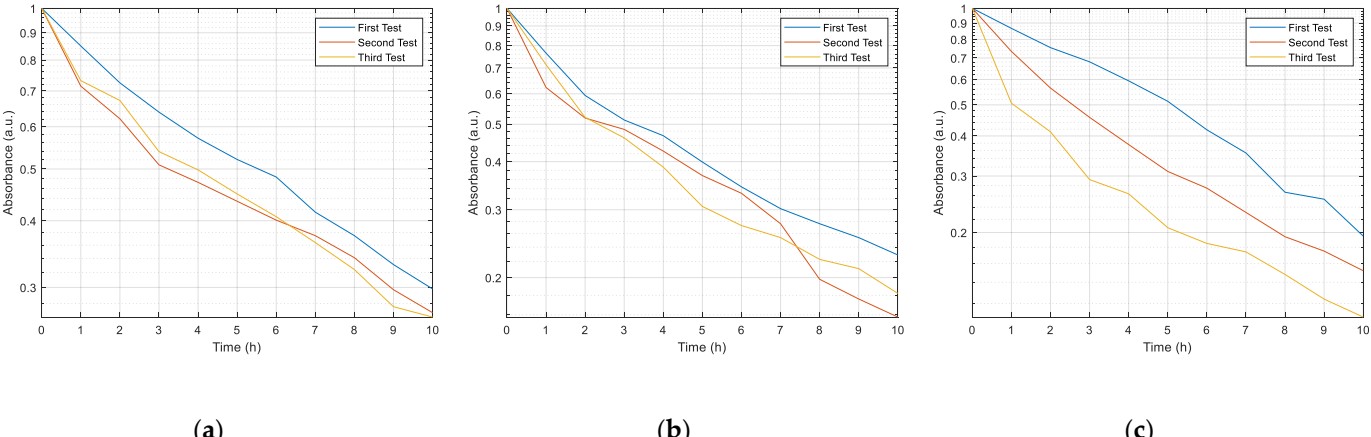

**Figure 11.** Photocatalytic reutilization of the different samples after three tests. (**a**) PAA + β-CD//TiO$_2$/PAA; (**b**) PAA + β-CD//TiO$_2$/PAA + WO$_3$; (**c**) PAA + β-CD//TiO$_2$/PAA + Fe$_2$O$_3$.

### 3.4. Electrochemical Measurements

All of the coated samples showed the same improvement in corrosion resistance compared to bare stainless steel, AISI 304. This meant that the coating process did not affect the surface despite the porosity of the coatings. Figure 12 compares the representative polarization curves corresponding to each coating system and the bare AISI 304. As can be seen, the coated samples showed corrosion potential shifted more towards noble values than AISI 304. Despite this trend, the changes were not statistically significant with respect to the steel substrate.

The same occurred with the passive current density. For all of the coated samples, the i$_{pass}$ ranged around $10^{-8}$ A/cm$^2$ no matter the coating system. Conversely, the pitting potential appeared to shift to higher values for the coatings containing two types of oxide particles. For both S2 and S3 coatings systems, the values for pitting potential showed statistically significant differences. Both coatings showed higher pitting potential resistance than the bare steel, pointing out an improved localized corrosion resistance of 50–100 mV in comparison to AISI304. These results are consistent with the observed thickening of

the fibers of the coatings, which reduced the pore size but also increased the thickness of the coatings.

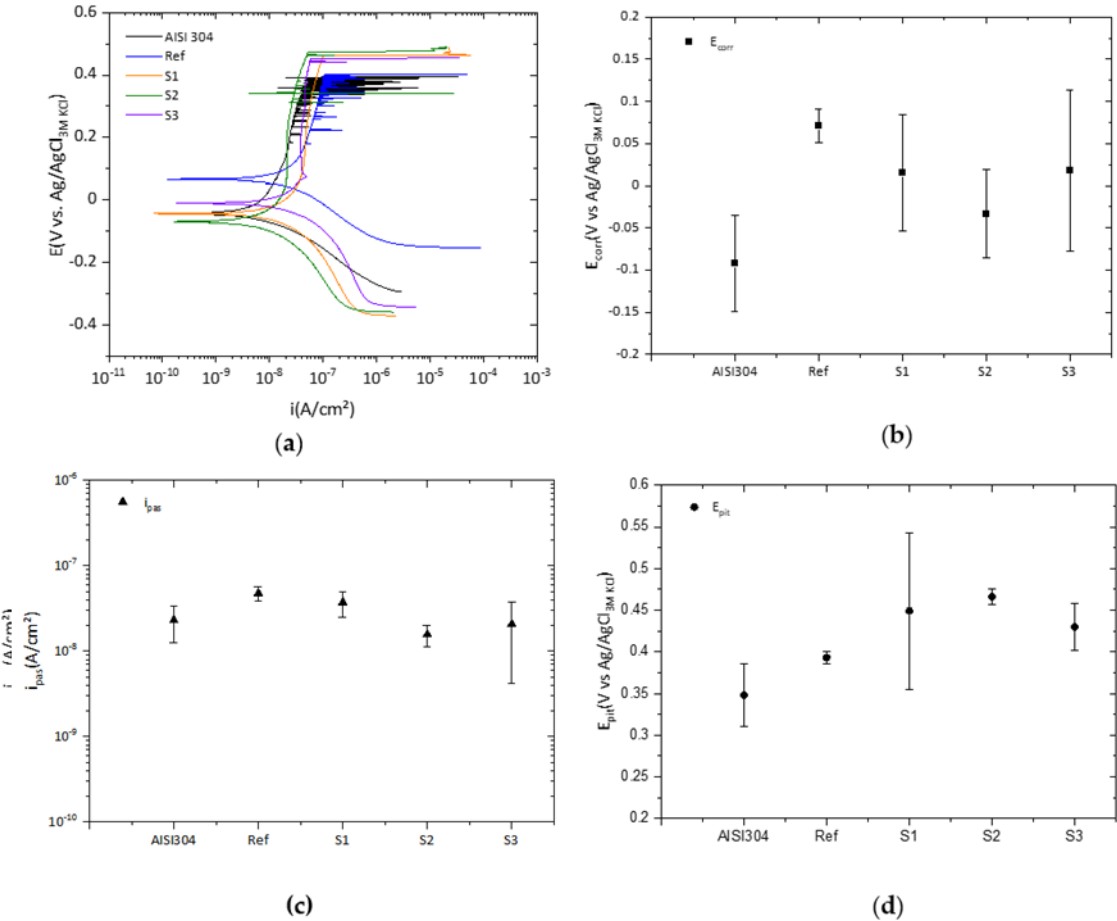

**Figure 12.** (**a**) Comparison of the representative potentiodynamic curves for the coated samples and AISI 304, and the average values of the main electrochemical parameters; (**b**) corrosion potential, $E_{corr}$; (**c**) passive current density, $i_{pass}$; (**d**) pitting potential, $E_{pit}$.

## 4. Conclusions

The combination of both electrospinning and layer-by-layer techniques to obtain a functional photocatalytic coating was successfully demonstrated. Metal oxide particles immobilized into the coating were able to change the superhydrophilic surface of the PAA+ β-CD into a less wettable surface. A total increase in water contact angle of almost 64° was achieved when $TiO_2$ and $WO_3$ particles were included in the coating. In addition, it was confirmed that once the layer-by-layer assembly technique was performed, the electrospun fiber diameter increased as a consequence of the multilayer deposition. Another aspect to mention was that the roughness of the deposition gradually decreased as the thickness of the coating increased. Thus, the typical porosity of electrospun surfaces was reduced due to the coarsening of the electrospun fiber diameter. Photocatalytic measurements showed the photodegradation of methylene blue. The combination of metal oxides improved the photocatalytic activity of the samples, leading to a total degradation of 85% when the combination of $TiO_2$ with $Fe_2O_3$ was included in the functional coating. Furthermore, the reutilization of these photocatalytic samples was also demonstrated by repeating the photocatalytic test over the same photodegraded area. Finally, the decrease in pore size in the functional coating improved the pitting potential of coated samples as metal oxides were added. Hence, the application of deposition to wastewater components made of stainless steel could be of great interest.

**Author Contributions:** Conceptualization, X.S., P.J.R. and R.J.R.; Data curation, X.S. and A.C.; Investigation, X.S., J.E., J.F.-P. and A.C.; Methodology, P.J.R. and R.J.R.; Project administration, P.J.R. and R.J.R.; Validation, P.J.R. and R.J.R.; Writing—original draft, X.S.; Writing—review and editing, P.J.R. and R.J.R. All authors have read and agreed to the published version of the manuscript.

**Funding:** This research was funded by the Government of Navarra—Department of Economic Development (PC019/020 ARGITU).

**Institutional Review Board Statement:** Not applicable.

**Informed Consent Statement:** Not applicable.

**Data Availability Statement:** Not applicable.

**Conflicts of Interest:** The authors declare no conflict of interest.

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
