# Peer review of "Design of Photocatalytic Functional Coatings Based on the Immobilization of Metal Oxide Particles by the Combination of Electrospinning and Layer-by-Layer Deposition Techniques"

_coatings, doi:10.3390/coatings12060862_

Round 1

Reviewer 1 Report

This study reports a novel photocatalytic functional coatings based on the immobilization of metal oxide particles by the combination of electrospinning and Layer-by-Layer deposition techniques, in order to obtain a functional, homogeneous and efficient photocatalytic coating. This study is novel and systematic. But some issues should be addressed before it can be published.

1、In the introduction, TiO2 and photocatalytic mechanism, which are widely known in related field, should be concised, and the research progress of the deposition techniques should be added.

2、The coating content of Fe2O3 and WO3 in the sample needs to be added.

3、The EDX results showed that the metal oxide distribution is not uniform. Will this affect the photocatalytic efficiency? And can this be avoided by adjusting the parameters of the deposition process?

4、The relationship between the fiber diameter and roughness of the samples and the photocatalytic efficiency needs to be explained.

5、The average coating thickness for S1 and S2 coating systems need tobe supplemented in Figure 8.

6、It is suggested to add the photocatalytic degradation efficiency under UV and visible light respectively.

7、"leading to nearly a complete degradation of the colorant (85%)" The statement is uncritical.

Author Response

We would like to thank to the anonymous reviewer for his/her positive comments. All the changes have been highlighted in yellow for a better understanding and localization of them, as it can be appreciated in the revised version of the manuscript. Finally, we hope that this new revised version of the manuscript can be published in the journal of Coatings. 

Reviewer 2 Report

The author fabricated the metal oxide composites photocatalysts by combining two different deposition methods. This work can be published after the author solve the following issues.

Some comments are given as follows for enhancing this work:

1. Please add a clear motivation in the Introduction part for the combination of TiO2 with other metal oxides such as Wo3 and Fe2O3.

2. Some related references based on photocatalysts should be cited: Advanced Functional Materials 31, 2021, 2104231; Advanced Materials Interfaces 8, 2021, 2001627.  

3. Generally, the size of the metal oxide will affect the catalytic performance of the catalyst. What is the size of the metal oxide particles in SEM images?

4. In EDX images, the author should point out which color corresponds to which element

5. Why S3-coating system present higher catalytic performance than S1- and S2-coating systems, please add the scientific reasons in the main text.

Author Response

(The authors gave the same response as above.)

Reviewer 3 Report

This is a fairly solid work, which should undoubtedly be recommended for publication, but after clarifying some incomprehensible points.

1.     The introduction is written quite clearly and all the important points are considered and conveyed to the readers. However,  because TiO2 is the main material for this work,  perhaps it is important to note that many such studies have been published in MDPI journals. Few of them are below:

Serga, V.; et al  Crystals 202111, 431. https://doi.org/10.3390/cryst11040431

Sun, X.; et al Nanomaterials 202212, 201. https://doi.org/10.3390/nano12020201

Kokorin, A.I.; et al . Catalysts 202010, 1022. https://doi.org/10.3390/catal10091022

Tsebriienko, T.; et al Crystals 2021, 11, 794. https://doi.org/10.3390/cryst11070794

2.     Paragraph 3.3 Photocatalytic Efficiency. Fig.9. It is clearly seen that the spectra in the figure consist of several components. Perhaps it makes sense to divide them into individual Gaussian components, then one could see how close and how different the processes are in these four cases. See also how individual bands change over time.

3.     Fig. 10. Is it possible to decompose the decay kinetics into individual exponential exponents?  This is a simple task, just change the vertical scale. See, as an example:

Fig. 6 and 7 and text in Karipbayev, Z.T. et al. Nucl. Instrum. Methods Phys. Res. B. Beam Interact. Mater. Atoms 2020, 479, 222–228.

Author Response

(The authors gave the same response as above.)

Round 2

Reviewer 1 Report

The manuscript has been well revised and was suggested to be accepted.

Reviewer 2 Report

The revised manuscript can be accepted.

Reviewer 3 Report

The authors have significantly improved the manuscript, constructively taking into account all the recommendations, so that the manuscript can be recommended for publication.